

# Cross-cultural adaptation and psychometric validation of a Chinese self-intermittent catheterization quality of life scale among patients with neurogenic bladder

Rong Tang[1] and Liqiong Zhou[2]

[1] Chengdu Medical College School of Nursing, Chengdu Medical College, Sichuan, Chengdu, China
[2] Department of Internal Medicine, South China Hospital of Shenzhen University, Shenzhen, Guangdong, China

## ABSTRACT

**Background:** Intermittent self-catheterization (ISC) is widely considered the gold standard for treating patients with neurogenic bladder (NB). Healthcare professionals and catheter users must optimize ISC care to improve patients' quality of life. To achieve this, the Intermittent Self-Catheterization Questionnaire (ISC-Q) is a valuable tool with clear and easy-to-understand items. However, this scale has yet to be adapted for use in China, and its reliability and validity need to be tested through cross-cultural adaptation.

**Objectives:** The objective of this study was to culturally adapt the ISC-Q and develop the Chinese version of the ISC-Q (C-ISC-Q), and assess its reliability and validity among patients with NB.

**Methods:** With the authorization of the ISC-Q author, the Beaton mode was used to adapt the scale cross-culturally. The study was conducted from June 2020 to June 2021 in three phases: committee-based translation from English to Chinese, Delphi expert consultations ($n = 5$) for cultural adaptation, and a cross-sectional study ($n = 149$) for validation.

**Results:** The C-ISC-Q has 24 items and four dimensions. The critical ratio of each item is over 3.0, and the correlation coefficient between each item and the scale's total score is above 0.4. The Cronbach's α value for the scale is 0.930, and for each dimension, it is between 0.870–0.92. The retest reliability for the scale is 0.894, and for each dimension is between 0.751–0.889. The content validity at the item level ranges from 0.8~1.0, and at the scale level is 0.9. The criterion-related validity of the scale was −0.708, and the correlation for each dimension was 0.329–0.624. The principal component analysis identified four common factors, with a cumulative contribution rate of 67.846%.

**Conclusions:** The C-ISC-Q is culturally sensitive, reliable, and valid to measure the quality of life for patients with NB. It can assist nurses and researchers in tailoring strategies to enhance the quality of life for patients with NB.

Corresponding author
Liqiong Zhou, 659788509@qq.com

# INTRODUCTION

Neurogenic bladder (NB) or neurogenic lower urinary tract dysfunction (NLUTD) refers to vesicourethral dysfunction resulting from damage to the central or peripheral nervous systems that control urination. This condition is characterized by symptoms such as urinary incontinence, urinary retention, frequent urination, and painful urination (*Kennelly et al., 2022*). Severe cases can lead to life-threatening complications such as renal failure, significantly impacting the patient's quality of life (*Panicker, 2020*).

The diagnosis and treatment of NB have become more sophisticated, with intermittent self-catheterization (ISC) being the preferred clinical treatment option (*Nambiar et al., 2022*). The procedure involves the insertion of a catheter into the bladder when the patient needs catheterization and removing it after emptying (*Wang et al., 2021*). The principle behind ISC is to reduce the bladder's residual urine volume, restore blood flow to the bladder wall, induce intermittent bladder expansion, and prevent continuous filling or expansion of the bladder. As per the guidelines, ISC not only effectively reduces the risk of urinary infections but also aids in the reconstruction of the reflective bladder, optimizes renal function, and significantly improves patients' overall quality of life (*Kinnear et al., 2020*).

The study highlights that the compliance of ISC patients is low, and while 60% of discharged patients can independently perform ISC, within 5 years, only 30% of patients can sustain it. The psychological issues NB patients face during ISC include anxiety, low self-confidence, and low acceptance (*Newman et al., 2020*; *Xiang et al., 2022*), while objective issues include discomfort during catheter insertion and inadequate public toilet facilities. These factors severely impact patient compliance, leading to recurrent urinary infections and significantly reducing their overall quality of life (*Gacci et al., 2022*).

It is essential to understand the objective problems and psychological state of NB patients during ISC, which can ensure the effective implementation of ISC and improve patients' quality of life (*Roberson et al., 2021*). Currently, the assessment tools for NB patients in China mainly focus on the psychological field or individual behaviour, such as the Intermittent Catheterization Acceptance Test (I-CAT), which evaluates the psychological acceptance of patients, and the Intermittent Catheterization Adherence Scale (ICAS), which assesses patient compliance. However, few studies have addressed specific assessment scales encompassing psychological state, objective problems, cognitive attitude, and other dimensions (*Elmelund, Klarskov & Biering-Sørensen, 2019*).

*Pinder et al. (2012)* developed the Intermittent Self-Catheterization Questionnaire (ISC-Q) in 2012, which consists of four dimensions of ease of use, convenience, discreetness, and psychological well-being with 24 items. It is mainly used to assess the quality of life of ISC patients (*Tate et al., 2020*), and has been translated into several versions, including Japan and Korean (*Zachariou et al., 2022*; *Scivoletto et al., 2017*; *Yeşil et al., 2020*), all of which show good reliability and validity. However, there is no Chinese

version (C-ISC-Q). The aim of this study was to translate, back-translate, and culturally adapt the ISC-Q in order to create a reliable and valid C-ISC-Q, which would provide a scientifically valid assessment tool for Chinese healthcare professionals.

## METHODS

### Setting and participants

This study was conducted in three tertiary hospitals located in Shenzhen, China. The subjects of the study were 149 NB patients who were hospitalized between June 2020 and June 2021. The snowball convenient sampling method was used to select the participants. According to the principle of reliability and validity test, the sample size is 5–10 times of the number of entries, ISC-Q has 24 entries, which requires 120–240 study subjects, and the minimum sample size is finally determined to be 144 cases, considering the 20% sample attrition rate.

Inclusion criteria for the study were: (i) NB patients who met the diagnostic criteria outlined in the Guidelines for the Diagnosis and Treatment of Neurogenic Gallbladder issued by the Chinese Urological Association (CUA); (ii) urine residue of ≥100 ml, requiring ISC; (iii) age ≥18 years old; and (iv) the ability to read and comprehend.

Exclusion criteria for the study were: (i) severe cardiac, brain, liver, or kidney dysfunction or malignant tumors; (ii) mental illness; and (iii) missing data of three or more items.

### Study procedures

The authors have permission to use this instrument from the copyright holders. This study followed the cross-cultural guidelines for scales proposed by Bitton (*Beaton et al., 2000*, *2007*), which involves six steps: initial translation, synthesis of translation, back translation, synthesis of back translation, review by a committee of experts, and pretesting (see Fig. 1 Cross-cultural steps of C-ISC-Q). The study was reported following the STARD standards introduced by *Pereira et al. (2019)*.

#### Translation of English scale into Chinese

Two Chinese researchers, one a physician from the rehabilitation department who has studied abroad in the United States, and the other a master of applied linguistics who has studied in the United Kingdom for 1 year, independently translated the scale into Chinese. They formed two separate translations: Translation Language 1 (TL1) and Translation Language 2 (TL2). The research team then integrated and revised TL1 and TL2 with the help of bilingual team members who grew up speaking Chinese and English. Finally, a comprehensive Chinese version of the scale was formed, referred to as Version I (*Sousa & Rojjanasrirat, 2011*).

#### Back translation of Chinese scale into English

Two English native researchers were invited to independently back-translate the Chinese scale. One is a rehabilitation doctor who has studied in China for 2 years, creating Back Translation Version 1 (B-TL1), and the other is an American English-speaking doctor, creating Back Translation Version 2 (B-TL2). The research team then compared B-TL1

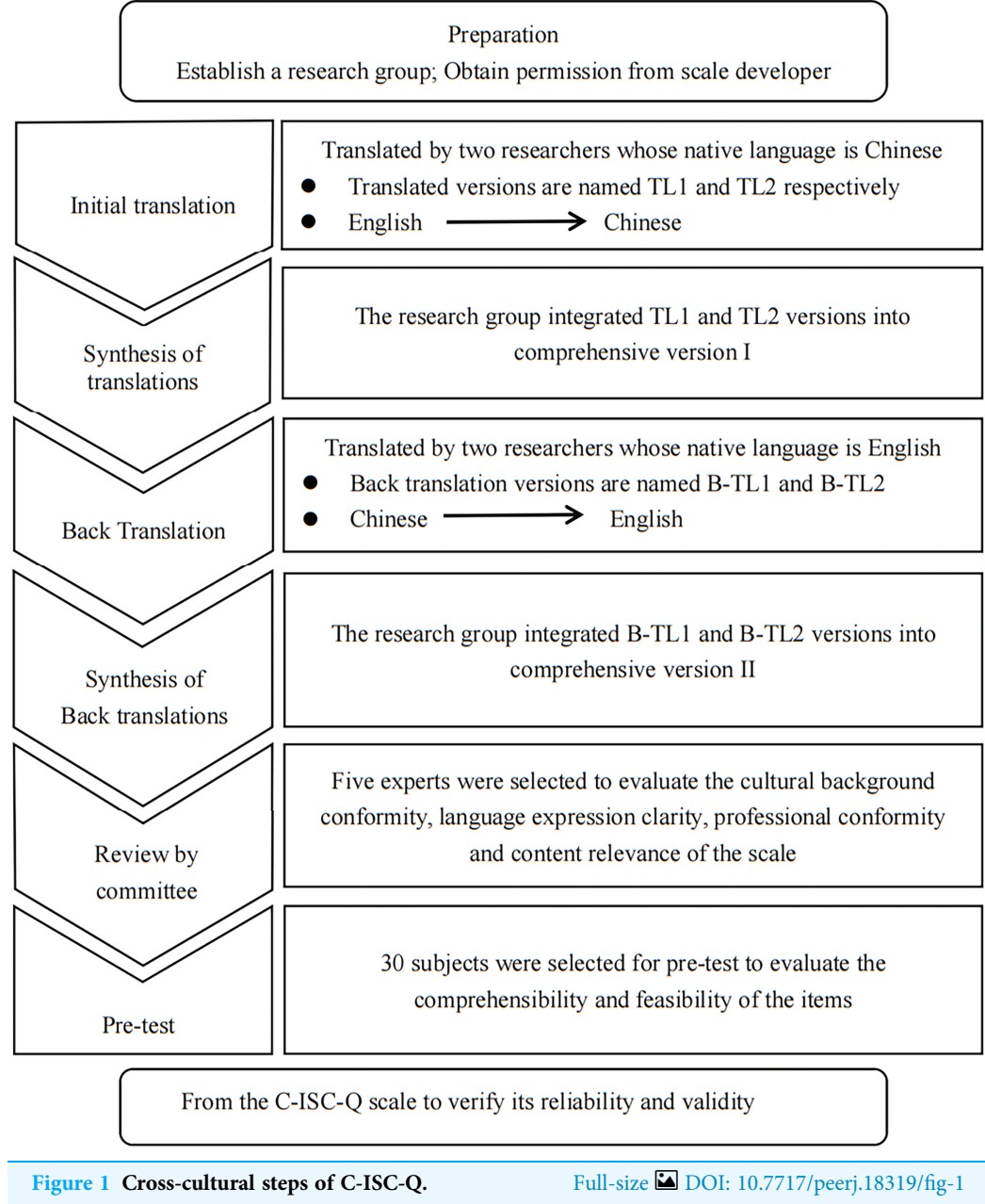

**Figure 1 Cross-cultural steps of C-ISC-Q.**

and B-TL2 with the original English scale and integrated the results to form a comprehensive English version of the scale, referred to as Version II.

*Cultural adaptation*

Five experts were selected to conduct cultural adaptation of the scale based on the following inclusion criteria: holding a bachelor's degree or above, having a deputy senior professional title or above, and having at least 10 years of work experience. The team comprised two clinical nursing experts, two chief physicians from the rehabilitation department, and one nursing college professor. Three experts held doctoral degrees, one

**Table 1 General information of experts (*n* = 5).**

| No. | Gender | Title | Major | Highest education | Familiarity | Age/ Working year | Judgment basis | | | |
| --- | --- | --- | --- | --- | --- | --- | --- | --- | --- | --- |
| | | | | | | | Theoretical knowledge | Practical experience | Reference | Subjective judgment |
| 1 | Female | Chief nurse | Nursing | Doctor | Very familiar | 60/42 | Large | Large | Large | Small |
| 2 | Female | Deputy chief nurse | Nursing | Doctor | Very familiar | 48/20 | Large | Large | Large | Medium |
| 3 | Female | Professor | Nursing | Undergraduate | Very familiar | 60/39 | Large | Large | Medium | Small |
| 4 | Female | Deputy chief physician | Clinical | Doctor | Very familiar | 42/11 | Large | Medium | Large | Large |
| 5 | Female | Chief physician | Clinical | Master | Familiar | 60/34 | Medium | Large | Medium | Small |

had a master's degree, and one held an undergraduate degree. Among them, three held senior professional titles, and two held deputy senior professional titles. Their working years ranged from 11 to 42 years, averaging 29.2 ± 13.22 years. The experts were asked to provide modification suggestions in four areas: cultural background conformity, language expression clarity, professional conformity, and content relevance of the scale. The Likert four scale was used to evaluate the suggestions, with a score of one indicating "very irrelevant," a score of two indicating "irrelevant," a score of three indicating "relevant," and a score of four indicating "very relevant." Details of the evaluation results can be found in Table 1.

### Pretest

To assess the comprehensibility and feasibility of the scale items and options, the research group used a convenient sampling method to select 30 subjects who met the inclusion criteria for the pretest. The pretest answer format was open-ended to encourage subjects to provide suggestions for modification.

## Data collection measures

For this study, the investigators used a general data questionnaire and the C-ISC-Q. Before conducting the survey, the researchers trained the investigators on the research purpose, methods, questionnaire collection, and precautions. The survey was conducted using both electronic and paper questionnaires. The first page of both types of questionnaires explained the study's purpose, significance, research methods, and confidentiality policy. Participants were required to read the contents on the first page and click "Confirm to participate in this study" to enter the questionnaire. If participants chose "Do not participate in this study," the answer process would end. The article questionnaire was distributed face-to-face, and participants responded on the spot before returning it. To ensure the quality of the questionnaire, the following inclusion criteria were set: (i) the number of missing values in the questionnaire was ≤3; (ii) the answer time was between 3 to 20 min, and (iii) the same mobile phone, computer, and IP address could only answer once.

## Statistical analysis

Microsoft Excel 2010 was used for data collation, while IBM SPSS 27.0 (SPSS ver. 27.0, Armonk, NY, USA) was used for data analysis. Continuous data was expressed as $\bar{X} \pm S$, counting data as frequency and percentage, and missing values were filled using the mean value of this item. Item analysis was conducted using the critical ratio (CR), and items with CR < 3.0 or a correlation coefficient between item score and total score of the scale < 0.4 were removed. The reliability of the scale was evaluated using Cronbach's α and retest reliability. An α coefficient > 0.7 indicated good internal consistency, while > 0.8 indicated very good internal consistency. Retest reliability was expressed by the correlation coefficient between the first test result and the retest result, with r > 0.4 indicating acceptability, and >0.75 indicating good reliability (*Hayase et al., 2022*). The scale's validity was evaluated using three indicators: content, criterion-related, and construct validity. Content validity included the item-level content validity index (I-CVI) and scale-level content validity index (S-CVI). Criterion-related validity was expressed by the correlation coefficient between SF-qualiveen and C-ISC-Q. Construct validity was measured by exploratory factor analysis. Data with a KMO > 0.7 and Bartlett coefficient $P < 0.01$ were suitable for factor analysis. Statistically significant differences were set as $P < 0.05$.

## Ethical considerations

This study received approval from the Ethics Review Committee of South China Hospital affiliated with Shenzhen University (HNLS20221209002-A). All participants are willing to participate in this study and provide written informed consent.

# RESULTS

## Results of cultural adaptation

Five experts were enlisted to facilitate the cultural adjustment. After consulting with them, the research group implemented eight modifications, with two focusing on cultural background (item 15.16), three on language expression (items 6, 17, and 24), two on professional conformity (item 7.20), and one on content relevance (item 5). Refer to Table 2 for the experts' recommendations and the outcomes of each modification.

## Pretest results

The study initially pretested 30 subjects who met the inclusion criteria. Based on the pretest results, modifications were made to five items related to language expression (items 3, 4, 10, 11, and 13). Refer to Table 2 for specifics. The final version of the C-ISC-Q was developed based on these modifications.

## Participant characteristics

A total of 155 questionnaires were distributed in this study, out of which 149 were valid, and six were invalid (three were repeated, and the response time of four questionnaires was <3 min). The questionnaire response rate was 96.13%. Refer to Fig. 2 for the flowchart depicting the inclusion of subjects. Of the 149 respondents, 98 were males, and 51 were

**Table 2** Modification of each item of C-ISC-Q and its CVI.

| No. | Item | N | I-CVI | S-CVI |
|-----|------|---|-------|-------|
| Q1 | It is easy to prepare my catheter for use each time I need it | 5 | 1 | 0.975 |
| Q2 | It is messy to prepare my catheter for use | 5 | 1 | |
| Q3 | It is easy to insert my catheter | 5 | 1 | |
| | **P:Problems in language expression** | | | |
| | *M:I think it is easier to insert catheter* | | | |
| Q4 | I find inserting the catheter is uncomfortable sometimes | 5 | 1 | |
| | **P:Word order** | | | |
| | *M:Sometimes inserting catheter makes me feel uncomfortable* | | | |
| Q5 | The design of the catheter makes it easy to insert | 5 | 1 | |
| | **C:The insertion position of the catheter is the urethra. It is recommended to add the "urethra"** | | | |
| | *M:The design of the catheter makes it easy to insert into the urethra* | | | |
| Q6 | The catheter is fiddly to use | 4 | 0.8 | |
| | **C:The word "complicated" is too written** | | | |
| | *M:I think it is inconvenient to use catheter* | | | |
| Q7 | The lubrication on the catheter makes it difficult to use | 5 | 1 | |
| | **C:Lubricant is located on the surface of urinary catheter, it is recommended to supplement** | | | |
| | *M:The lubricant on the surface of the catheter makes it difficult to use* | | | |
| Q8 | I feel confident in my ability to use my catheter | 5 | 1 | |
| Q9 | Storage of catheters at home is inconvenient | 5 | 1 | |
| Q10 | Taking enough catheters for a weekend away is very inconvenient | 5 | 1 | |
| | **P:Problems in language expression** | | | |
| | *M:It is not convenient to carry enough catheter when going out on weekends* | | | |
| Q11 | Taking enough catheters for a 2-week holiday is very inconvenient | 5 | 1 | |
| | **P:Problems in language expression** | | | |
| | *M:It is not convenient to carry enough catheters for two weeks of vacation* | | | |
| Q12 | Disposal of my catheter is inconvenient when away from home | 4 | 0.8 | |
| Q13 | I find it easy to carry enough catheters around with me on a day-to-day basis | 5 | 1 | |
| | **P:Problems in language expression** | | | |
| | *M:It is easy to carry enough catheter around me every day* | | | |
| Q14 | I find it easy to dispose of my catheter when I am away from home | 5 | 1 | |
| Q15 | My catheter is discreet | 5 | 1 | |
| | **C:The expression is not concise enough** | | | |
| | *M:My catheter is hidden* | | | |
| Q16 | I can use my catheter discreetly when I am away from home | 5 | 1 | |
| | **C:Pay attention to the concealment of the catheter.** | | | |
| | *M:When going out, I can use my catheter more covertly* | | | |
| Q17 | I can easily dispose of my catheter without it being obvious to people | 5 | 1 | |
| | **C:The expression is not concise enough** | | | |
| | *M:I can easily handle the catheter when others are not paying attention* | | | |
| Q18 | My catheter allows me to feel confident when away from home | 5 | 1 | |
| Q19 | I am self-conscious about my need to self-catherize | 5 | 1 | |

| Table 2 (continued) | | | | |
|---|---|---|---|---|
| **No.** | **Item** | **N** | **I-CVI** | **S-CVI** |
| Q20 | I would feel embarrassed if people saw my catheter in its packet | 5 | 1 | |
| | **C:The position of the catheter and urine bag should be emphasized during the catheterization** | | | |
| | *M:I will feel embarrassed because others see my catheter or urine bag* | | | |
| Q21 | My need to use a catheter sometimes makes me feel embarrassed | 4 | 0.8 | |
| Q22 | I worry that my catheter doesn't always empty my bladder fully | 5 | 1 | |
| Q23 | My need to use catheters stops me from visiting friends and family as often as I would like | 5 | 1 | |
| Q24 | I worry about the risk of long-term problems from using my catheter | 5 | 1 | |
| | **C:"Risk" is derogatory and too written** | | | |
| | *M:I am worried that using a catheter will cause long-term problems* | | | |

**Note:**
P, Pretest comments; M (modification), modification results; C (comment), expert opinions; N, the number of experts who agree to the modification results. The item without C/M means no expert opinion has yet to be modified.

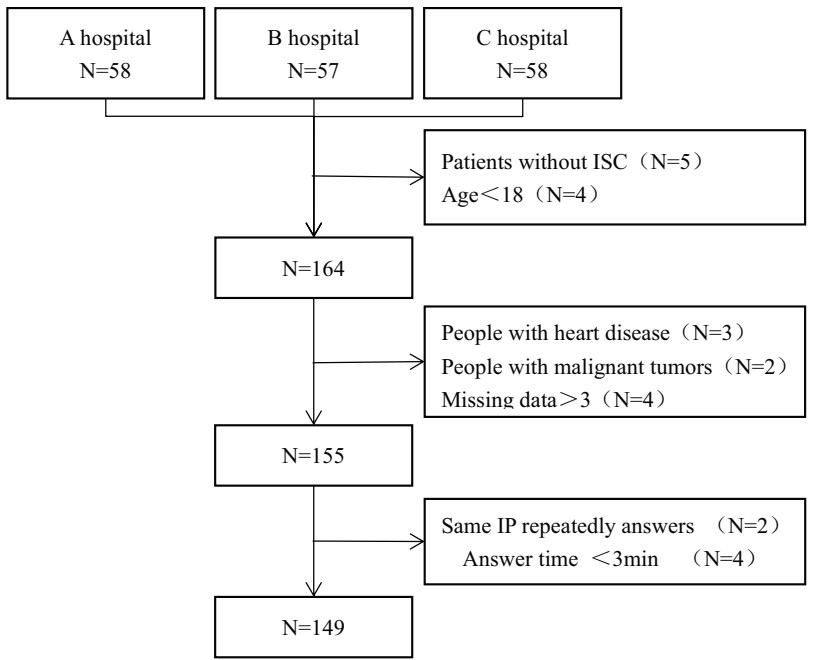

**Figure 2 The flow chart for inclusion of objects.** 

females, with an average age of 35.45 ± 10.74 years. Refer to Table 3 for further details regarding the subjects' general information.

## Item analysis results

The study results indicated that each item of the C-ISC-Q had a CR of 5.991−0.92 ($P < 0.01$) and a correlation coefficient between 0.510−0.702 ($P < 0.01$) with the scale. As a result, no items were deemed necessary to be removed, and the scale was considered to have good discrimination. Please see Table 4 for the item analysis results.

**Table 3 The general information of subjects (_n_ = 149).**

| Content | | Number | Percentage (%) |
|---|---|---|---|
| Gender | Male | 98 | 65.77 |
| | Female | 51 | 34.23 |
| Age | ≤30 years old | 54 | 36.24 |
| | 31~50 years old | 74 | 49.67 |
| | ≥50 years old | 21 | 14.09 |
| Residence | Rural areas | 69 | 46.31 |
| | City | 80 | 53.69 |
| Marital status | Unmarried | 36 | 24.16 |
| | Married | 104 | 69.80 |
| | Divorce | 5 | 3.36 |
| | Bereavement | 4 | 2.68 |
| Education | Junior high school and below | 28 | 18.79 |
| | High school | 43 | 28.86 |
| | Junior college or above | 78 | 52.35 |
| Care mode | Live alone | 40 | 26.85 |
| | Spouse care | 79 | 53.02 |
| | Child care | 10 | 6.71 |
| | Parental care | 14 | 9.40 |
| | Sibling care | 6 | 4.03 |
| Length of illness | <1 year | 36 | 24.16 |
| | 1~5 year | 49 | 32.89 |
| | ≥5 year | 64 | 42.95 |
| Cause of illness | Spinal cord injury | 63 | 42.28 |
| | Cerebrovascular accident | 31 | 20.81 |
| | Basal ganglia lesions | 6 | 4.03 |
| | Intervertebral disc disease | 13 | 8.72 |
| | Spinal stenosis | 9 | 6.04 |
| | Spina bifida | 10 | 6.72 |
| | Brain tumor | 9 | 6.04 |
| | Multiple sclerosis | 4 | 2.68 |
| | Other | 4 | 2.68 |
| Have you ever experienced any complications | Yes | 78 | 52.35 |
| | No | 71 | 47.65 |

## Reliability results

The study findings indicate that Cronbach's α of the C-ISC-Q was 0.930, while Cronbach's α of each dimension ranged from 0.870 to 0.925. To evaluate the retest reliability, the research group selected 30 subjects using a random number table and conducted a follow-up investigation two weeks later. The results revealed that the retest reliability was 0.894, with each dimension ranging from 0.751 to 0.889. For more information, refer to Table 5.

**Table 4 Item analysis of Chinese ISC-Q.**

| Item No. | High grouping | Low grouping | CR | r |
|---|---|---|---|---|
| Q1 | 3.17 ± 0.74 | 1.49 ± 1.19 | 7.71** | 0.666** |
| Q2 | 3.34 ± 0.57 | 1.66 ± 1.26 | 7.80** | 0.664** |
| Q3 | 3.39 ± 0.77 | 1.24 ± 0.99 | 10.92** | 0.702** |
| Q4 | 3.12 ± 0.84 | 1.37 ± 0.99 | 8.63** | 0.655** |
| Q5 | 3.32 ± 0.79 | 1.32 ± 1.01 | 9.99** | 0.690** |
| Q6 | 3.07 ±0.82 | 1.27 ± 1.16 | 8.13** | 0.658** |
| Q7 | 3.46 ± 0.50 | 1.66 ± 1.26 | 8.53** | 0.652** |
| Q8 | 3.22 ± 0.72 | 1.49 ± 1.27 | 7.60** | 0.575** |
| Q9 | 3.51 ± 0.51 | 1.66 ± 1.20 | 9.14** | 0.688** |
| Q10 | 3.32 ± 0.65 | 1.80 ± 1.08 | 7.70** | 0.551** |
| Q11 | 3.24 ± 0.70 | 1.73 ± 1.10 | 7.45** | 0.577** |
| Q12 | 3.29 ± 0.51 | 1.93 ± 1.13 | 7.07** | 0.609** |
| Q13 | 3.39 ± 0.70 | 1.51 ± 1.33 | 8.02** | 0.653** |
| Q14 | 3.29 ± 0.68 | 1.78 ± 1.35 | 6.40** | 0.535** |
| Q15 | 3.07 ± 0.85 | 1.71 ± 1.19 | 5.99** | 0.510** |
| Q16 | 3.27 ± 0.84 | 1.68 ± 1.33 | 6.46** | 0.547** |
| Q17 | 3.20 ± 0.75 | 1.61 ± 1.12 | 7.56** | 0.583** |
| Q18 | 3.49 ± 0.55 | 1.80 ± 1.35 | 7.41** | 0.595** |
| Q19 | 3.41± 0.71 | 1.78 ± 1.31 | 7.02** | 0.600** |
| Q20 | 3.41 ± 0.63 | 1.85 ± 1.06 | 8.09** | 0.619** |
| Q21 | 3.37 ± 0.70 | 1.90 ± 1.24 | 6.58** | 0.615** |
| Q22 | 3.37 ± 0.73 | 1.68 ± 1.15 | 7.90** | 0.646** |
| Q23 | 3.20 ± 0.68 | 1.51 ± 0.90 | 9.57** | 0.647** |
| Q24 | 3.29 ± 0.51 | 1.98 ± 1.29 | 6.06** | 0.607** |

Note:
** Indicates $P < 0.01$

**Table 5 Reliability results of C-ISC-Q.**

| | First test score | Retest score | Cronbach's α | Retest reliability |
|---|---|---|---|---|
| Total scale | 66.53 ± 17.87 | 61.77 ± 26.80 | 0.930 | 0.894** |
| Ease of use | 64.83 ± 24.49 | 54.17 ± 29.60 | 0.925 | 0.889** |
| Convenience | 68.29 ± 22.98 | 59.31 ± 26.46 | 0.870 | 0.751** |
| Discreetness | 63.39 ± 25.30 | 70.28 ± 26.35 | 0.905 | 0.843** |
| Psychological well-being | 69.60 ± 22.01 | 61.38 ± 17.97 | 0.887 | 0.754** |

Note:
** Indicates $P < 0.01$.

## Validity results

The study's content validity results revealed that the I-CVI of the C-ISC-Q ranged from 0.800 to 1.000, and the S-CVI was 0.975, indicating good content validity. Refer to Table 2 for specifics.

**Table 6 The criterion-related validity results of C-ISC-Q.**

| | Score | Ease of use | Convenience | Discreetness | Psychological well-being | Total scale |
|---|---|---|---|---|---|---|
| Bother | 2.36 ± 1.13 | −0.389** | −0.328** | −0.413** | −0.323** | −0.484** |
| Constrains | 2.20 ± 1.10 | −0.484** | −0.465** | −0.432** | −0.507** | −0.624** |
| Fear | 2.67 ± 1.07 | −0.185* | −0.184* | −0.387** | −0.224* | −0.329** |
| Feeling | 2.41 ± 1.15 | −0.393** | −0.321** | −0.262** | −0.245** | −0.406** |
| Total scale | 2.41 ± 0.73 | −0.559** | −0.499** | −0.571** | −0.498** | −0.708** |

Notes:
* Indicates $P < 0.05$.
** Indicates $P < 0.01$.

Regarding criterion-related validity, the SF-qualiveen score was 2.41 ± 0.73, while the C-ISC-Q score was 66.53 ± 17.87. The correlation coefficient between the two scales was −0.708 ($P < 0.001$), and the absolute value of the correlation coefficient for each dimension was 0.184–0.507. These findings suggest that the C-ISC-Q has good criterion-related validity, but the correlation of each dimension is low. Please refer to Table 6 for more information.

To assess the scale's construct validity, the research group conducted an exploratory factor analysis. The analysis results indicated a KMO of 0.917, and an X2 value of 2199.572, $P < 0.001$, signifying the suitability of factor analysis. The principal component analysis and maximum orthogonal rotation method were utilized to extract four common factors, which had an initial eigenvalue of >1 and accounted for 38.616%, 13.185%, 9.442%, and 6.603%, respectively. The cumulative variance contribution rate was 67.846%. Furthermore, the factor load of each item ranged from 0.701 to 0.842, and the factor attribution of each item was consistent with the original scale. Please refer to Table 7 for further details.

# DISCUSSION

The results of this study suggest that the C-ISC-Q scale can be used to assess the quality of life of patients with intermittent catheterization of neurogenic bladder. The scale has high internal consistency and retest reliability, good content validity and calibration correlation validity. The scale has a moderate number of entries and an easy-to-understand presentation, and is rated on a 5-point Likert scale (0–4), with higher scores indicating higher quality of life. It was concluded that the localized C-ISC-Q scale can be promoted as a proprietary tool for assessing the quality of life of patients with intermittent catheterization.

During the cross-cultural adaptation process, this study followed the BEATON theory, which consists of six steps. In step five, only five experts were included, which may be considered a small number of experts. However, the BEATON theory does not specify the exact number of experts to be included. According to the literature, the number of experts in similar studies ranges from 5 to 12. Therefore, this study included five experts in the consultation process and made eight modifications to develop C-ISC-Q.

**Table 7 The exploratory factor analysis results of C-ISC-Q.**

| Dimension | Item | Component | | | |
|---|---|---|---|---|---|
| | | 1 | 2 | 3 | 4 |
| Ease of use | It is easy to prepare my catheter for use each time I need it | 0.729 | 0.142 | 0.19 | 0.163 |
| | It is messy to prepare my catheter for use | 0.807 | 0.105 | 0.152 | 0.13 |
| | I think it is easier to insert catheter | 0.761 | 0.15 | 0.196 | 0.186 |
| | Sometimes inserting catheter makes me feel uncomfortable | 0.714 | 0.187 | 0.21 | 0.076 |
| | The design of the catheter makes it easy to insert into the urethra | 0.739 | 0.069 | 0.271 | 0.194 |
| | I think it is inconvenient to use catheter | 0.782 | 0.122 | 0.227 | 0.032 |
| | The lubricant on the surface of the catheter makes it difficult to use | 0.842 | 0.083 | 0.125 | 0.106 |
| | I feel confident in my ability to use my catheter | 0.804 | −0.004 | 0.065 | 0.144 |
| Convenience | Storage of catheters at home is inconvenient | 0.252 | 0.218 | 0.283 | 0.753 |
| | It is not convenient to carry enough catheter when going out on weekends | 0.122 | 0.149 | 0.2 | 0.789 |
| | It is not convenient to carry enough catheters for two weeks of vacation | 0.21 | 0.164 | 0.135 | 0.790 |
| | Disposal of my catheter is inconvenient when away from home | 0.182 | 0.138 | 0.273 | 0.775 |
| Discreetness | It is easy to carry enough catheter around me every day | 0.059 | 0.779 | 0.31 | 0.214 |
| | I find it easy to dispose of my catheter when I am away from home | 0.1 | 0.802 | 0.096 | 0.088 |
| | My catheter is hidden | 0.096 | 0.790 | −0.002 | 0.17 |
| | When going out, I can use my catheter more covertly | 0.19 | 0.823 | 0.06 | −0.002 |
| | I can easily handle the catheter when others are not paying attention | 0.087 | 0.751 | 0.178 | 0.198 |
| | My catheter allows me to feel confident when away from home | 0.122 | 0.818 | 0.18 | 0.081 |
| Psychological well-being | I am self-conscious about my need to self-catherize | 0.109 | 0.164 | 0.799 | 0.14 |
| | I will feel embarrassed because others see my catheter or urine bag | 0.17 | 0.185 | 0.725 | 0.17 |
| | My need to use a catheter sometimes makes me feel embarrassed | 0.228 | 0.128 | 0.749 | 0.104 |
| | I worry that my catheter doesn't always empty my bladder fully | 0.225 | 0.171 | 0.755 | 0.133 |
| | My need to use catheters stops me from visiting friends and family as often as I would like | 0.293 | 0.06 | 0.701 | 0.241 |
| | I am worried that using a catheter will cause long-term problems | 0.218 | 0.064 | 0.718 | 0.223 |

The discrimination of each item is mostly reflected by CR (*Rechenchosky et al., 2022*). In this study, the CR of each item indicates that there are statistical differences between the high and low groups, that is, each item has good discrimination, so no item is deleted.

Therefore, each item can effectively identify the difference in quality of life with different scores. In addition, this study analyzes the common variability of each item and the scale. The results showed that the content measured by the items and the content reflected by the scale had a high correlation; that is, the common variation was large, indicating that the homogeneity between the items was good. Few studies have reported CR in ISC-Q intercultural adaptation, so this study could not find relevant studies reported for comparison.

Internal consistency and stability are important indicators of scale reliability. This study uses Cronbach's α to reflect internal consistency and retest reliability to reflect stability. The Cronbach's α value is 0.930 for the scale and 0.870–0.92 for each dimension, indicating that the scale has excellent internal consistency (*Pinder et al., 2012*), consistent with the source scale. Moreover, the retest reliability of each dimension and the scale is higher than 0.80, consistent with the retest reliability of the source scale, indicating that C-ISC-Q has good stability. In conclusion, the C-ISC-Q scale has good internal consistency and stability, suggesting that it has good reliability.

Validity reflects the degree to which a measuring tool accurately reflects the expected concept (*Menegol et al., 2022*). In this study, content validity was evaluated by five experts. The item-level content validity index (I-CVI) ranged from 0.8 to 1.0. The scale-level content validity index (S-CVI) was 0.975, indicating that the measured content had a high correlation with the target measurement content. The original version of ISC-Q did not report content validity, so there is no basis for comparison. Moreover, this study used SF-qualiveen as the criterion (*Rong et al., 2022*). The criterion-related validity was acceptable, but the correlation of each dimension was low. In 2012, Pinder's study reported that the correlation coefficient between ISC-Q and Qualiveen was 0.64, and the correlation coefficient of each dimension was 0.27~0.77, similar to the results of this study (*Kako et al., 2022*). This may be because SF-qualiveen focuses on the psychological field of quality of life, while ISC-Q focuses more on the problem of dysfunction. Although both scales evaluate the quality of life, they focus on different aspects, resulting in lower dimension correlation. Finally, the construct validity results showed that the number of common factors was consistent with the source scale. The cumulative contribution rate of the four factors was 67.85%, higher than the 48.90% of the source scale. The possible reason for the difference is that the sample size in this study was smaller (149 patients) than that in the source study (306 patients) (*Pinder et al., 2012*). Therefore, further verification is needed.

The study has several limitations. The sample size used in this study is relatively limited as only neurogenic bladder patients in Shenzhen were selected, and the investigation was not conducted in other regions using larger sample sizes. This may result in some sampling bias, and further exploration is needed to assess the applicability of the scale in the different areas. Moreover, confirmatory factor analysis was not conducted due to the inclusion of only 149 subjects in the study, which is insufficient for the requirement of confirmatory factor analysis of having a sample size of more than 100 cases (*Fatoye et al., 2022*). As a result, the study has certain limitations.

## CONCLUSION

In conclusion, the ISC-Q has been widely used since its early development. In this study, the C-ISC-Q scale was translated into Chinese and was found to have reasonable content, clear expression, and consistency with the cultural background. It takes only 5–10 min to complete and has good reliability and validity, making it a useful tool for medical staff to understand the dysfunction and psychological problems of patients with neurogenic bladder during ISC. The C-ISC-Q can effectively evaluate the quality of life of ISC patients and play a significant role in monitoring, evaluating, and comparing patients' quality of life with different causes.

## ABBREVIATION LIST

| | |
|---|---|
| **ISC** | Intermittent self-catheterization |
| **NB** | Neurogenic bladder |
| **ISC-Q** | Intermittent self-catheterization questionnaire |
| **C-ISC-Q** | Chinese version of the intermittent self-catheterization questionnaire |
| **NLUTD** | Neurogenic lower urinary tract dysfunction |
| **I-CAT** | Intermittent catheterization acceptance test |
| **ICAS** | Intermittent catheterization adherence scale |
| **CUA** | Chinese urological association |
| **TL1** | Translation language 1 |
| **TL2** | Translation language 2 |
| **B-TL1** | Back translation version 1 |
| **B-TL2** | Back translation version 2 |
| **CR** | Critical ratio |
| **I-CVI** | Item level content validity index |
| **S-CVI** | Scale level content validity index |

## ACKNOWLEDGEMENTS

The researchers would like to express the gratitude to the authors of the source scale for authorizing the sinicization of the ISC-Q. They would also like to thank the experts for their careful guidance during the consultation process. In addition, they would also like to express their gratitude to the research subjects of neurogenic bladder patients included in the data collection process for their support and cooperation.

### Funding

The authors received no funding for this work.

### Competing Interests

The authors declare that they have no competing interests.

## Author Contributions

- Rong Tang conceived and designed the experiments, performed the experiments, analyzed the data, prepared figures and/or tables, and approved the final draft.
- Liqiong Zhou conceived and designed the experiments, performed the experiments, authored or reviewed drafts of the article, and approved the final draft.

## Human Ethics

The following information was supplied relating to ethical approvals (*i.e.*, approving body and any reference numbers):

South China Hospital of Shenzhen University approval to carry out the study within its facilities.

## Field Study Permissions

The following information was supplied relating to field study approvals (*i.e.*, approving body and any reference numbers):

This study received approval from the South China Hospital affiliated with Shenzhen University.

## Data Availability

The raw measurements are available in the Supplemental File.

## Supplemental Information

Supplemental information for this article can be found online at http://dx.doi.org/10.7717/peerj.18319#supplemental-information.

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
