# Peer review of "Cross-cultural adaptation and psychometric validation of a Chinese self-intermittent catheterization quality of life scale among patients with neurogenic bladder"

_PeerJ, doi:10.7717/peerj.18319_

## Round 0.1 · original submission · Major Revisions

Thank you for submitting your manuscript titled "Chinese Self-Intermittent Catheterization Quality of Life Scale." The reviewers have provided valuable feedback, and I have carefully considered their comments. While your study demonstrates promising outcomes in culturally adapting and validating the Intermittent Self-Catheterization Questionnaire (ISC-Q) for the Chinese population, there are several critical areas that need to be addressed before the manuscript can be considered for publication.

A significant concern raised by the reviewers, which I also share, is the relatively small sample size of 149 participants. This limitation may affect the generalizability and robustness of your findings. Unless there is a strong, well-justified rationale for using this sample size, I recommend expanding the sample to include a larger and more diverse population. This will strengthen the validity of your study and enhance the reliability of the C-ISC-Q as a tool for broader application.

Reviewers have questioned why the study focused solely on patients with Neurogenic Bladder (NB). It would be beneficial to provide a clear and detailed rationale for this decision in the manuscript. Clarifying why only NB cases were included will help readers understand the scope and applicability of your findings.

The demographic table currently lacks detailed characteristics of the patients, particularly the cause of NB. Including this information is crucial as it provides context and allows for a better understanding of the study population. Please add these characteristics to your demographic table.

Reviewers have suggested that the main findings of the study should be presented at the beginning of the discussion rather than being included in the introduction. This reorganization will help emphasize the significance of your results and provide a clearer narrative flow.
Additionally, consider exploring the responsiveness of the C-ISC-Q in detecting changes in quality of life over time in future research.

Reviewer 1 ·

Basic reporting

The study titled "Chinese Self-Intermittent Catheterization Quality of Life scale" aims to culturally adapt and validate the Intermittent Self-Catheterization Questionnaire (ISC-Q) for use in China. The research evaluates the reliability and validity of the Chinese version of the ISC-Q (C-ISC-Q) among patients with Neurogenic Bladder (NB). The methodology used in this study follows a systematic approach. The Beaton model was employed to cross-culturally adapt the ISC-Q, ensuring the questionnaire's appropriateness for the Chinese population. The study was conducted in three phases, including committee-based translation, Delphi expert consultations, and a cross-sectional study for validation. The results of the study demonstrate promising outcomes for the C-ISC-Q. The questionnaire consists of 24 items divided into four dimensions, and each item shows a critical ratio above 3.0. The correlation coefficient between each item and the total score of the scale is also positive, indicating good internal consistency. The Cronbach's value for the overall scale and each dimension suggests high reliability. Furthermore, the retest reliability indicates good stability over time. The content validity of the C-ISC-Q is supported by high validity scores at both the item and scale levels. Additionally, the criterion-related validity shows a negative correlation, indicating that the scale effectively measures the intended construct. The principal component analysis reveals four common factors, explaining a substantial proportion of the variance.
In conclusion, the study successfully culturally adapts and validates the C-ISC-Q for measuring the quality of life in Chinese patients with NB. The questionnaire demonstrates good reliability, validity, and content sensitivity. The findings suggest that the C-ISC-Q can be a valuable tool for healthcare professionals and researchers in tailoring strategies to enhance the quality of life for patients with NB.

However, it is important to consider some limitations of the study. The sample size of 149 participants may be relatively small, and it would be beneficial to include a larger and more diverse sample to strengthen the generalizability of the findings.
Additionally, further research could explore the responsiveness of the C-ISC-Q in detecting changes in quality of life over time.
My comments are as below:
Why did the authors include only NB cases?
In the demographic table, it is suggested to add other characteristics of patients such as the cause of NB.
The first paragraph of the discussion should be belonged to the main finding of work.

Experimental design

It is mentioned in the above box.

Validity of the findings

It is mentioned in the above box.

Reviewer 2 ·

Basic reporting

It is well written.

Experimental design

My Comments are as follow:
Why did the authors focused solely on Neurogenic Bladder (NB) cases, and it would be helpful to clarify the rationale behind this decision.
Please add the characteristics, such as the cause of NB of the included subjects.
It is recommended to reorganize the discussion section, starting with the main findings of the study rather than including them in the introduction.

Validity of the findings

It is confirmed.

Reviewer 3 ·

Basic reporting

Clear, objective language, current and relevant references. Describes the importance of the instrument within the theme.

Experimental design

Methodology follows scientific rigor with all procedures for cultural adaptation and translation well described.

Validity of the findings

The results and conclusion show relevance and are associated with the proposed objectives.

Additional comments

Study within the relevant theme, the introduction addresses the subject in a clear and objective way, with current and relevant references from academic literature; describes the instrument in a way that is easy to understand.
The methodology is well outlined within the expected precepts for translation and cultural adaptation

---

## Round 0.2 · accepted · Accept

Dear Authors,

Thank you for submitting the revised version of your manuscript. We appreciate your efforts in addressing the previous comments. There is, however, a minor adjustment that still needs to be made. This can be corrected during the proof stage, so no further revision is required at this point.

Best regards,

Reviewer 2 ·

Basic reporting

The authors made necessary changes in the revised version of the manuscript. The only comment that should be addressed is the replacement of "neurogenic bladder" with "neurogenic lower urinary tract dysfunction (NLUTD)".

Experimental design

The authors made necessary changes in the revised version of the manuscript.

Validity of the findings

The authors made necessary changes in the revised version of the manuscript.

Additional comments

The authors made necessary changes in the revised version of the manuscript.